## RESEARCH ARTICLE

# Comprehensive comparative analysis of the effects of temperature on the Notch signaling response *in vivo*

**Nimmy S. John[1,2], Kah Seng Tang[1,2,3], Michelle A. Urman[1,2] and ChangHwan Lee[1,2,*]**

## ABSTRACT

Temperature is a critical factor that modulates cellular metabolism and stem cell regulation. Despite extensive studies, the influence of temperature on stem cell regulation via Notch signaling has been limited to studies relying on studies that involve indirect readouts to Notch activation. This study systematically analyzes the effects of temperature on the Notch signaling transcriptional response at the chromosomal, cellular, and tissue levels. Using complementary direct Notch readouts, we demonstrate that Notch activation remains largely unchanged across temperatures, suggesting the presence of temperature-compensatory mechanisms that maintain robust Notch activation. Notch transcriptional activity readouts, however, increased with temperature, indicating that elevated temperatures may enhance Notch transcriptional activity at the chromosomal level. These findings provide a comprehensive framework for understanding effects of temperature and offer new insights into the regulation of Notch signaling in stem cell biology.

KEY WORDS: *Caenorhabditis elegans* gonad, Notch signaling, Temperature, Transcriptional regulation, Spatial pattern analysis, *sygl-1*, Gradient

## INTRODUCTION

Temperature is a crucial regulator of developmental timing and physiological homeostasis in both ectothermic and endothermic organisms (Nomura et al., 2022). Fluctuations in temperature can lead to short-term and long-term cellular changes, such as alterations in cell composition and proteins folding (Knapp and Huang, 2022). Studies have shown that temperature modulates cellular metabolism, influencing the rate of gene expression essential for organismal development (Nomura et al., 2022; Begasse et al., 2015; Kuntz and Eisen, 2014). For example, elevated temperatures can accelerate enzymatic reaction rates and protein turnover, while cooler conditions can impair membrane fluidity and mitochondrial efficiency, altering energy balance (Hazel, 1995; Somero, 1995). Temperature shifts also influence cell-cycle dynamics, such as progression through G1/S and mitotic checkpoints, thus affecting tissue growth and regeneration

(Begasse et al., 2015; Rieder and Cole, 2002). This plays a vital role in stem cell regulation by affecting the regulation of signaling pathways that govern the viability, proliferation, and differentiation of stem cells (Gandara and Drummond-Barbosa, 2022).

Notch signaling is a conserved cell signaling pathway that operates via a common mechanism across metazoans, playing a pivotal role in regulating cell fate decisions and tissue patterning (Albert Hubbard and Schedl, 2019; Artavanis-Tsakonas et al., 1999; Crittenden et al., 2019; Greenwald et al., 1983; Wilkinson et al., 1994). In *Drosophila*, Notch signaling remains stable across temperature fluctuations due to temperature-dependent compensatory mechanisms (Nomura et al., 2022; Chong et al., 2018). In contrast, in chick amniote brains, short-term hypothermia has been shown to enhance Notch activity and suppress neurogenesis in neural progenitor cells (Nomura et al., 2022). However, most of these studies rely on indirect readouts, such as GFP driven by Notch responsive promoters, which can under- or overestimate the immediate effects of temperature changes in the native context due to the long half-lives of these reporters. This highlighted the need for more direct and sensitive assays to assess how temperature modulates Notch activation and signaling dynamics.

Here, we focus on Notch activation in the *Caenorhabditis elegans* germline, where it directly drives transcription of two Notch target genes, *sygl-1* and *lst-1*, to maintain a pool of 30-75 germline stem cells (GSCs) at the distal end of the gonad (Lee et al., 2016; Austin and Kimble, 1987; Chen et al., 2020). Temperature modulates multiple aspects of *C. elegans* physiology, including developmental speed, locomotion patterns, egg-laying rates, and chemosensory behaviors (Zhang et al., 2015; Ikeda et al., 2021; Parida et al., 2014; Li et al., 2024; Adachi et al., 2008). Notably, an approximate 5°C increase in growth temperature accelerates developmental timing by about 50%, whereas lowering temperature is associated with extended lifespan (Lee et al., 2016; Zhang et al., 2015; Gómez-Orte et al., 2017). Despite these broad effects, how temperature influences Notch activation and its functional consequences in the germline remains poorly understood. Here, we address this gap by using direct readouts of Notch activity to systematically analyze the effects of commonly used worm growth temperatures (15°C, 20°C, 22.5°C, and 25°C) on Notch-dependent transcriptional response.

Here, we used single-molecule fluorescent *in situ* hybridization (smFISH) to perform a comprehensive comparative analysis of Notch-induced transcription across temperatures (15°C, 20°C, 22.5°C and 25°C). The spatial pattern and overall level of Notch transcriptional response remain largely unchanged across temperatures, suggesting the presence of temperature-compensatory mechanisms that maintain robust Notch activation. Individual active transcription site (ATS) intensities and cytoplasmic mRNA counts, however, increased with temperature, indicating that elevated temperatures may enhance Notch transcriptional activity at the chromosomal level.

[1]Department of Biological Sciences, University at Albany, State University of New York, Albany, NY, 12222, USA. [2]The RNA Institute, University at Albany, State University of New York, Albany, NY, 12222, USA. [3]Department of Biochemistry, Stony Brook University, 100 Nicolls Rd, Stony Brook, NY 11794, USA.

*Author for correspondence (chlee@albany.edu)

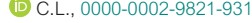 C.L., 0000-0002-9821-9312

Biology Open

## RESULTS

To systematically analyze the effects of temperature on Notch signaling response in the *C. elegans* germline, we performed smFISH to visualize transcripts of a major Notch target, *sygl-1*, in young-adult wild-type (N2) worms grown at different temperatures (15°C, 20°C, 22.5°C and 25°C) (Fig. 1A). *sygl-1* smFISH assays have been established as direct readouts of Notch-induced transcriptional activation and its spatial pattern (Crittenden et al., 2019; Lee et al., 2016; Urman et al., 2023; John et al., 2025 preprint; Kershner et al., 2014; Lynch et al., 2022). These readouts include *sygl-1* nascent transcripts at ATS, reflecting chromosomal-level Notch activity, and mature cytoplasmic mRNAs, estimating cellular-level activity (Crittenden et al., 2019; Lee et al., 2016; Urman et al., 2023; Kershner et al., 2014; Lynch et al., 2022; Shin et al., 2017; Sorensen et al., 2020). Previous work established that *sygl-1* smFISH provides direct readouts of Notch transcriptional activation and activity using two probe sets, each targeting the exon or intron regions of the Notch target gene *sygl-1* (Crittenden et al., 2019; Lee et al., 2016; Urman et al., 2023; John et al., 2025 preprint; Kershner et al., 2014; Lynch et al., 2022). Intron probes highlight nascent transcripts at the ATS, while exon probes detect both nascent transcripts (bright foci in Fig. 1A, 'Intron') and cytoplasmic mRNAs (dimmer foci in Fig. 1A, 'Exon') (Crittenden et al., 2019; Lee et al., 2016; Urman et al., 2023; John et al., 2025 preprint; Kershner et al., 2014; Lynch et al., 2022). When combined with DAPI-stained nuclei, these signals enable the unambiguous identification and confirmation of ATS (Crittenden et al., 2019; Lee et al., 2016; Urman et al., 2023; John et al., 2025 preprint; Kershner et al., 2014; Lynch et al., 2022). Notch activity at 20°C was consistent with previously reported measurements across chromosomal, cellular, and tissue levels (Lee et al., 2016; Urman et al., 2023; John et al., 2025 preprint; Lynch et al., 2022) (Fig. 1, 20°C).

### Notch-induced transcriptional activation remains unchanged across different temperatures

To compare tissue-level Notch-induced transcriptional activation across temperatures, we scored the number of cells containing *sygl-1* ATS and the extent of *sygl-1* ATS along the gonadal axis, which reflects the overall Notch transcriptional response and the size of Notch-responsive germ cell pool, respectively (Fig. 1B,C). The number of cells containing *sygl-1* ATS remained consistent across all temperatures, indicating that Notch activation is unaffected by temperature at the tissue level (Fig. 1B). The extent of ATS was slightly increased at 15°C (Fig. 1C). To assess Notch activation at the cellular level, we quantified *sygl-1* ATS per ATS-containing cell within two regions: 0-30 µm from the distal end, where most GSCs reside, and 0-60 µm, which encompasses roughly two-thirds of the progenitor zone (PZ) (Fig. 1D,E, Fig. S1A-C). Neither the number nor the composition of ATS per cell varied from 15-22.5°C, whereas both parameters showed a slight increase at 25°C, confirming that Notch activation remains robust across temperatures at both the cellular and tissue levels (Fig. 1D,E, Fig. S1A,B). Supporting this, the distributions of germ cell nuclear sizes, indicative of cell cycle stage, were largely unchanged across temperatures, except for a slight increase at higher temperatures, consistent with accelerated cell cycle progression (Fig. 1F, Fig. S1D). In line with this, we observed a small, though not statistically significant, increase in mitotically dividing cells at elevated temperatures (Fig. 1G), consistent with previous reports of accelerated cell cycles, which in turn allows more cells containing 3-4 ATS to occur at 25°C (Begasse et al., 2015; Richards et al.,

2013). Altogether, these results demonstrate that Notch-induced transcriptional activation remains unchanged at cellular and tissue levels across physiological temperature ranges.

### Transcriptional activation of a Notch-independent gene, *let-858*, increases with temperature

We next asked whether the temperature-independent consistency in transcriptional activation is a unique feature of Notch target genes, or a broader property shared with Notch-independent genes. To address this, we performed smFISH targeting *let-858* (Fig. 2A), a gene whose expression is independent of Notch signaling in germ cells (Lee et al., 2016; Urman et al., 2023; John et al., 2025 preprint; Ferdous et al., 2003; Kelly et al., 1997). We focused our analysis on the first 30 µm of the distal gonad, where the majority of GSCs reside (Lee et al., 2016; Cinquin et al., 2010) and quantified the number of *let-858* ATS (Fig. 2B). In contrast to the *sygl-1* ATS, the number of *let-858* ATS increased with temperature (Figs 1B-D and 2B). A similar, though statistically insignificant, increase was observed in the number of nuclei (Fig. 2C) and in the number of *let-858* ATS per cell at elevated temperatures (Fig. 2D). Additionally, *let-858* ATS-containing cells were distributed uniformly across the distal germline (0-60 µm from the distal end) and across all temperatures (Fig. 2E,F). These results suggest that temperature-independent transcriptional consistency is specific to Notch-induced transcriptions and does not extend to all actively transcribed genes.

### Notch-induced *sygl-1* transcriptional activity increases with temperature

Although Notch-induced transcriptional activation, which reflects the number of GSCs or chromosomes responding to Notch signaling, remains unchanged across temperatures (Fig. 1), we asked whether transcriptional activity, defined as the amount of RNA produced, is affected by temperature. To assess *sygl-1* transcriptional activity at the chromosomal level, we measured individual *sygl-1* ATS intensities at different temperatures and observed a gradual increase with rising temperature (Fig. 3A). Similarly, the summed *sygl-1* ATS intensity per nucleus, a proxy for cellular-level transcriptional activity, also increased with temperature (Fig. 3B). This trend extended to the tissue-level as both the total number of *sygl-1* mRNA within the first 60 µm of the distal gonad (approximately two-thirds of the progenitor zone, PZ) and the average number of *sygl-1* mRNAs per cell increased with temperature (Fig. 3C,D). This trend persisted when analysis was restricted to the first 30 µm of the distal gonad, corresponding to the typical length of the germline stem cell (GSC) pool (Lee et al., 2016) (Fig. S1E,F). The PZ size, an estimate of gametogenesis capacity, also expanded at higher temperatures, although the trend did not precisely mirror changes in ATS intensities or mRNA levels (Fig. 3E). Together, these results indicate that while Notch-induced transcriptional activation is buffered against temperature changes, Notch activity increases with temperature at both chromosomal and cellular levels.

### The spatial distribution of Notch-induced *sygl-1* transcription is unaffected by temperature changes

Notch-induced transcriptional activation occurs in a steep gradient within the GSC pool at the distal gonad, which plays a crucial role in germline polarization and GSC maintenance (Lee et al., 2016; Urman et al., 2023; John et al., 2025 preprint). The spatial pattern of *sygl-1* ATS has also been established as a reliable indicator of the Notch-responsive GSC pool (Lee et al., 2016; Urman et al., 2023; John et al., 2025 preprint; Cinquin et al., 2010) (Fig. 4A, red-dashed lines). To determine whether temperature influences this graded

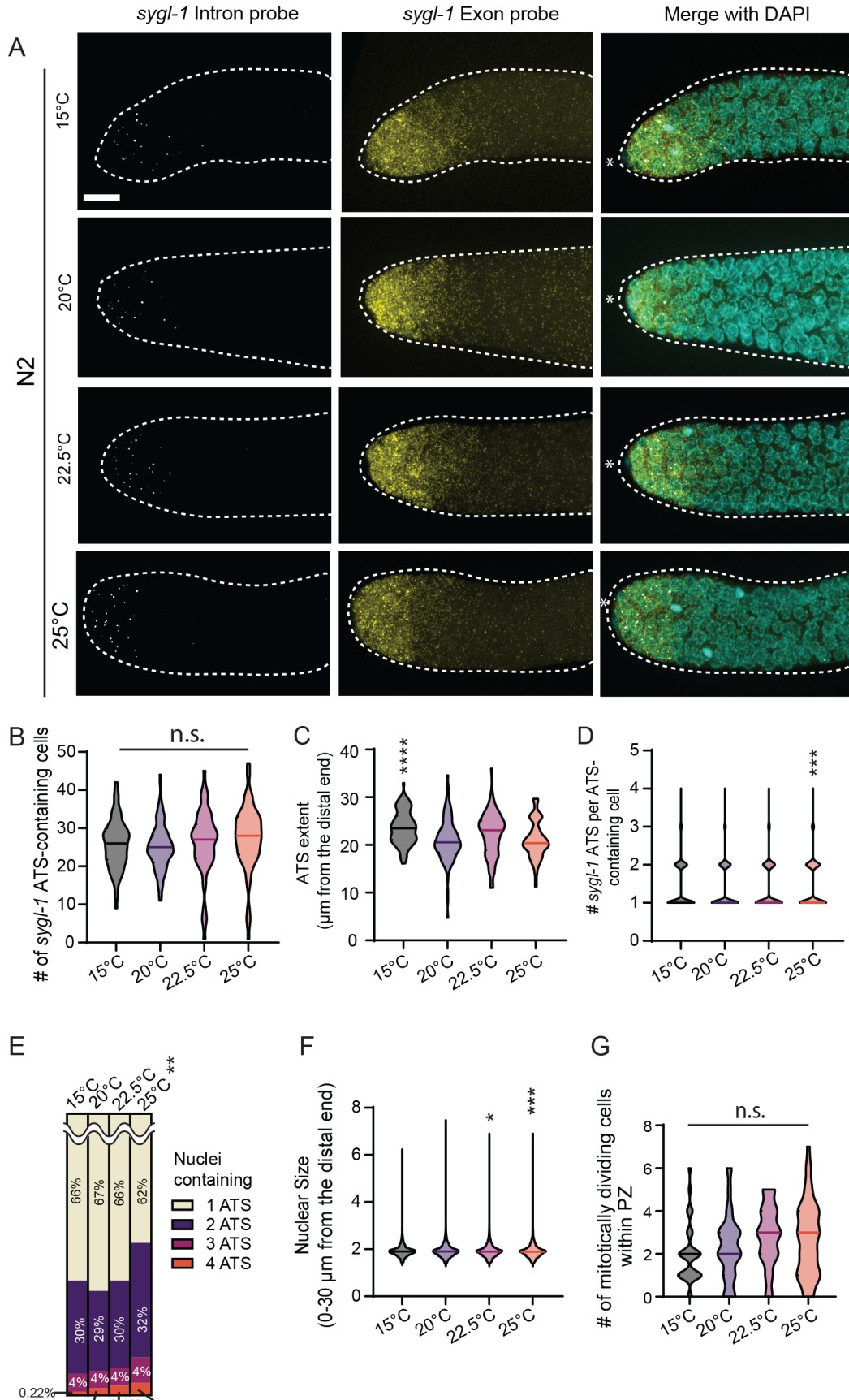

**Fig. 1.** See next page for legend.

**Fig. 1. Overall changes in *sygl-1* transcriptional response as temperature increases.** (A) Z projected *sygl-1* smFISH images that are representative of the *sygl-1* Notch response as temperature increased (15°C, 20°C, 22.5°C and 25°C). (B) # of *sygl-1* ATS-containing cells were plotted as a population for each temperature. (C) *sygl-1* ATS Extents were plotted as a population for each temperature. Sample sizes for temperatures, 15°C, 20°C, 22.5°C and 25°C were as follows: *n*=35, 56, 36, and 47 gonads, respectively. (D) The number of *sygl-1* ATS per ATS-containing cells was plotted as a population for each temperature. (E) Percentages of 1-4 *sygl-1* ATS distribution were plotted as a population for each temperature. Chi Square test performed for the statistical comparison between populations against the 20°C control. **P<0.01. (B,D-F) Sample sizes for temperatures, 15°C, 20°C, 22.5°C and 25°C were as follows: *n*=86, 118, 63, and 84 gonads, respectively. (F) Nuclear sizes via radius within 30 μm were plotted as a population for each temperature. Sample sizes for temperatures, 15°C, 20°C, 22.5°C and 25°C were as follows: *n*=6091, 10,330, 4899 and 6782 nuclei, respectively. (G) Mitotic cell counts were plotted as a population for each temperature. Sample sizes for temperatures, 15°C, 20°C, 22.5°C and 25°C were as follows: *n*=35, 56, 36, and 45 gonads, respectively. (A) Scale bar: 10 μm, white asterisks represent the distal end of the germline. In this work, the horizontal lines within all of the violin plots represent the median. (B-D,F,G) For this and all the following plots, statistical pairwise comparison was conducted on the experimental temperatures (15°C, 22.5°C and 25°C) against the control temperature (20°C) using two-tailed *t*-test, and *P*-values are denoted as follows *P<0.05; **P<0.01; ***P<0.001; ****P<0.0001; n.s., non-significant.

Notch response pattern and alters GSC pool size, we analyzed the spatial patterns of *sygl-1* ATS and mRNAs across temperature (Fig. 4). Specifically, we quantified the percentage of germ cells containing *sygl-1* ATS as a function of distance from the distal end of the gonad, which reflects the probability of Notch activation along the gonadal axis (Fig. 4A). Across all temperatures, the sygl-1 ATS gradient and the inferred GSC pool size remained largely unchanged, indicating that the spatial pattern of Notch activation is also buffered against temperature changes (Fig. 4A, red-dashed lines). This temperature-independent spatial pattern was also evident when analyzing the number of *sygl-1* ATS or number per cell and the percentage of germ cells with *sygl-1* mRNAs above the basal level (∼5 *sygl-1* mRNAs per cell) (Fig. 4B-D). These results indicate that the graded pattern of Notch-induced transcription and the size of the GSC pool are preserved across a range of physiological temperatures.

## DISCUSSION

This study systematically investigates how physiological temperatures affect Notch-induced transcription at chromosomal, cellular, and tissue levels in the *C. elegans* germline. Using direct, quantitative Notch transcriptional readouts, including *sygl-1* ATS and mRNAs, we show that the probability and spatial pattern of Notch activation remain largely invariant across physiological temperatures, with only slight increases at higher temperatures (15°C, 20°C, 22.5°C, and 25°C). In contrast, the transcription of the Notch-independent gene *let-858* increases with temperature (Fig. 2), consistent with previous reports showing temperature-sensitive regulation of transgene expression (Kelly and Fire, 1998; Couteau et al., 2002; Camacho et al., 2018). These findings suggest the presence of a buffering mechanism that maintains consistent Notch-induced transcriptional activation, likely to preserve GSC pool size and function under varying environmental conditions and ensure robust gametogenesis.

Despite stable Notch activation patterns, the transcriptional activity of *sygl-1*, as measured by individual ATS intensities and total mRNA per cell, increased with temperature (Fig. 3A-D). This enhanced Notch transcriptional activity correlates with expansion of

the PZ (Fig. 3E), which may ultimately impact fertility and progeny size. We speculate that increased activity or abundance of transcriptional regulators, such as the DNA-binding protein LAG-1/CSL or the Notch intracellular domain (NICD), may underlie this temperature-associated boost in *sygl-1* transcription. Further studies should assess whether temperature modulates the levels, binding dynamics, or nuclear accessibility of these key transcriptional components.

Notably, the temperature-buffering mechanism appears to regulate Notch transcriptional activation (i.e. the probability of Notch signaling response; Fig. 4A,B), but not the extent of transcriptional activity (i.e. the amount of RNA produced once activated; Fig. 4C,D). This distinction implies that buffering occurs upstream of target gene activation, potentially at the level of Notch ligand-receptor interactions. We speculate that temperature may influence the abundance or stability of Notch ligands (e.g. LAG-2/DSL) or modulate proteolytic processing of the GLP-1 receptor to release NICD. Elucidating how these molecular steps are insulated from temperature changes will be critical for understanding the robustness of Notch signaling in dynamic environments. Together, our findings reveal that while Notch activation is buffered against temperature changes, its transcriptional activity and output are more responsive, providing a layered regulatory architecture. This dual mode of control may allow for stable GSC regulation while enabling physiological flexibility in response to changing conditions.

## MATERIALS AND METHODS
### Nematode strains used in this study:

| Strain name | Genotype |
| --- | --- |
| N2 | Wild-type strain of *C. elegans* |

### Nematode culture
All strains were maintained at 20°C as previously described (Brenner, 1974). The wild type was N2 Bristol. For the smFISH experiments, all strains were synchronized via hypochlorite treatment and cultured on OP50-seeded NGM plates until the appropriate day of adulthood within each temperature.

| Temperature | Days from plated L1s |
| --- | --- |
| 15°C | 5 |
| 20°C | 3 |
| 22.5°C | ∼2.5 (54 h from L1) |
| 25°C | 2 (48 h from L1) |

### smFISH
smFISH for *sygl-1* and *let-858* were performed as previously described (Lee et al., 2016, 2017; Urman et al., 2023; John et al., 2025 preprint). Synchronized L1 larvae were grown on OP50 until day 1 of adulthood within their respective temperatures as described in the Nematode culture table above (John et al., 2025 preprint; Kershner et al., 2014). Briefly, the synchronized *C. elegans* of N2 for each experimental set were washed off plates with 2-3 ml non-RNase free 1X PBS+0.1% Tween-20 (PBST) and were collected on the 60 mm Petri dish cover. An additional 2-3 ml of non-RNase free PBST was added, and the worms were dissected to extrude the gonads in PBST with 0.25 mM levamisole added. The dissected samples were fixed with 3.7% formaldehyde in 1X PBS with 0.1% Tween-20 at room temperature (RT) for 30 min, with rotation. Samples were spun down at 2000 RPM for 1 min unless noted otherwise. After fixation, samples were permeabilized with the permeabilization buffer for 10 min at room temperature with rotation. The samples were then washed twice with RNase free PBST, resuspended in 70% ethanol, and stored overnight at 4°C.

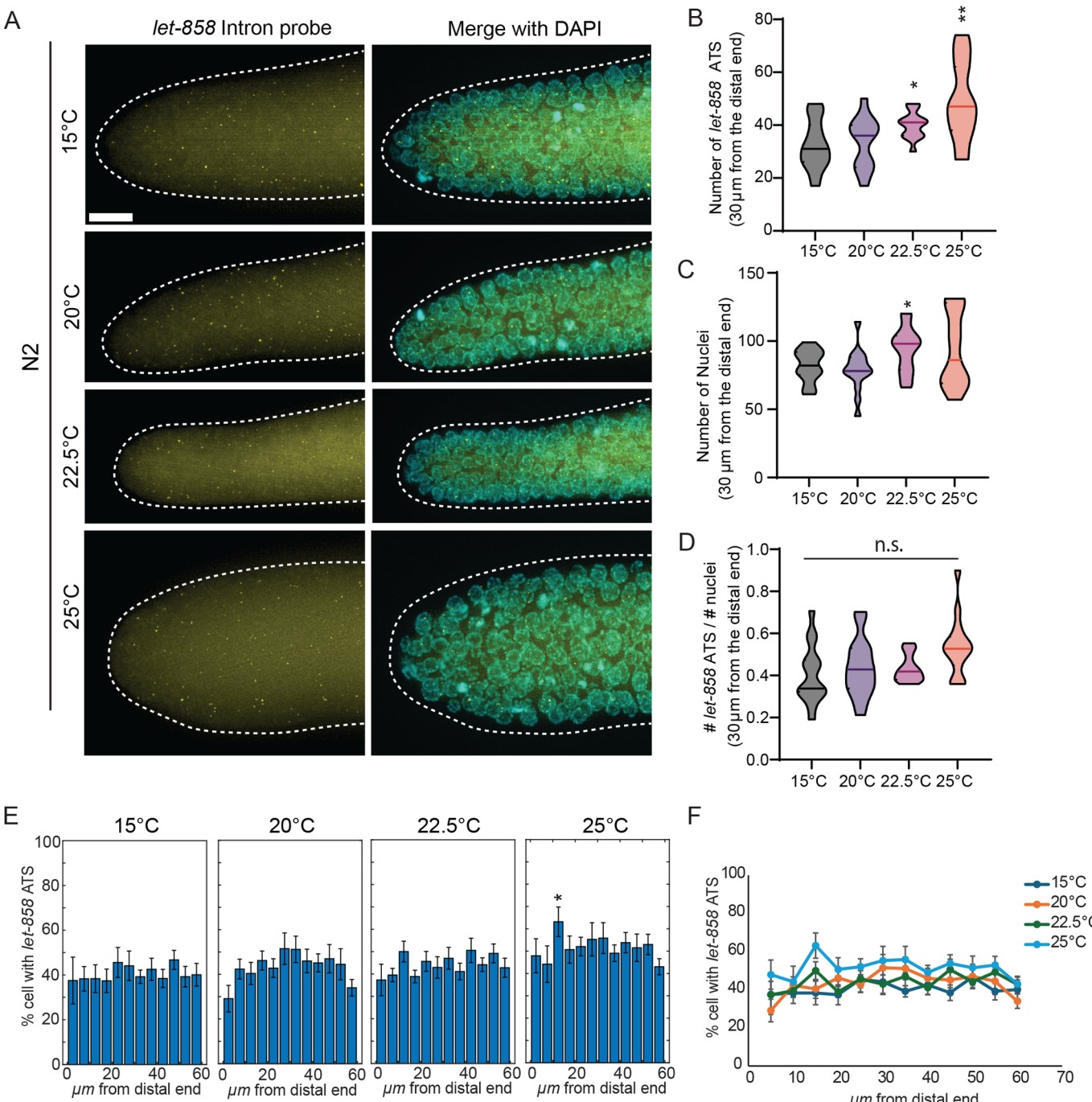

**Fig. 2. Effects of temperature on the *let-858* transcriptional response.** (A) Z projected representative *let-858* smFISH images as temperature increases. (B) Number of *let-858* ATS within 30 µm from the distal end was plotted as a population for each temperature. Sample sizes for all temperatures were *n*=15. (C) Germ cell nuclei were counted within 30 µm of the distal end of the gonad at each temperature. (D) The ratio of *let-858* ATS (from B) to nuclei (from C) was calculated for each gonad at each temperature. (E) The percentage of germ cells containing at least one *let-858* ATS was plotted as a function of distance from the distal end in 5 µm intervals, for each temperature. Temperatures are indicated above each plot (*n*=15 gonads per condition). (F) Data from E were converted to ridgeline plots and overlaid to facilitate comparison across temperatures. *P<0.05; **P<0.01; n.s., non significant by two-tailed *t*-test.

Custom Stellaris FISH probes (Biosearch Technologies, Inc., Petaluma, CA, USA) were designed against the exon and intron regions of *sygl-1* and the intron regions of *let-858* as described previously (Lee et al., 2016; Urman et al., 2023). Ethanol was removed and samples were incubated in 1 ml of wash buffer for 5 min at room temperature. Gonads were hybridized with 1 µl of each of the *sygl-1* probes (6.25 µM) or *let-858* probes (6.25 µM) in hybridization buffer for 24 h at 37°C with rotation. After probe addition, samples were kept in the dark for all incubations and washes. Samples were rinsed once with wash buffer at room temperature,

then incubated in wash buffer for 30 min at room temperature with rotation. The DNA was then labeled by incubation in smFISH wash buffer containing 1 mg/ml diamidinophenylindole (DAPI) for 30 min at room temperature followed by two short washes with smFISH wash buffer. Finally, samples were resuspended in 10-12 µl Antifade Prolong Gold mounting medium (Life Technologies Corporation, Carlsbad, CA, USA) and mounted on glass slides, which were then cured for 48 h. To analyze the smFISH images, we used customized MATLAB codes as previously described (Lee et al., 2016, 2017; John and Lee, 2025). To minimize

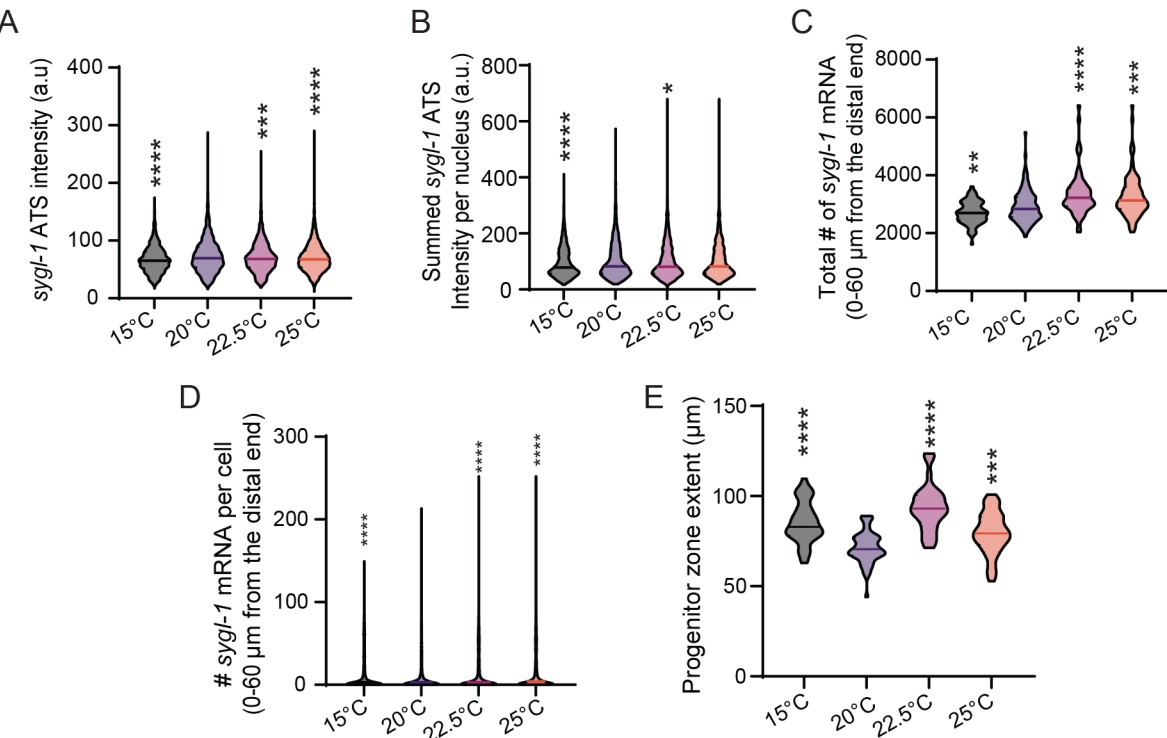

**Fig. 3. Overall changes in the Notch-dependent generation of cytoplasmic mRNA.** (A) *sygl-1* ATS intensity was plotted as a population for each temperature. Sample sizes for temperatures, 15°C, 20°C, 22.5°C and 25°C were as follows: *n*=3256, 4372, 2467 and 3416 nuclei, respectively. (B) Summed *sygl-1* ATS intensity was plotted as a population for each temperature. Sample sizes for temperatures, 15°C, 20°C, 22.5°C and 25°C were as follows: *n*=2223, 2954, 1654, and 2305 nuclei, respectively. (C) Total number of *sygl-1* mRNA within 60 μm from the distal end was plotted as a population for each temperature. Sample sizes for temperatures, 15°C, 20°C, 22.5°C and 25°C were as follows: *n*=86, 118, 63 and 84 gonads, respectively. (D) Number of *sygl-1* mRNA per cell within 60 μm from the distal end was plotted as a population for each temperature. Sample sizes for the temperatures, 15°C, 20°C, 22.5°C and 25°C were as follows: *n*=15198, 27271, 12676 and 16896 nuclei, respectively. (E) Progenitor zone extents were plotted as a population for each temperature. Sample sizes for temperatures, 15°C, 20°C, 22.5°C and 25°C were as follows: *n*=35, 56, 36, and 45 gonads, respectively. *$P<0.05$; **$P<0.01$; ***$P<0.001$; ****$P<0.0001$ by two-tailed *t*-test.

detection errors from photobleaching or heterogeneous background across 3D image stacks, MATLAB codes implement multiple normalization steps using background signals both inside and outside the germline and corresponding nucleus in each focal plane as previously established (John and Lee, 2025). For statistical analyses, we first performed one-way ANOVA to assess overall differences among multiple groups compared together (reported by *F*-values), followed by pairwise two-tailed *t*-tests to determine statistical significance between specific conditions (reported by *P*-values). Statistical significance is indicated with asterisks, as detailed in the figure legends.

**Microscopy setup and image acquisition**
Gonads were imaged completely (depth >15 μm) with a Z-step size of 0.3 μm using a Leica DMi8 Widefield Microscope that is equipped with a THUNDER Imaging system and computational clearing methods that are provided in the Leica Application Suite X (LAS X) acquisition software (Leica Microsystems Inc., Buffalo Grove, IL, USA) as previously described (Urman et al., 2023; John et al., 2025 preprint). All imaging was done with LED8 light sources, sequentially through the channels in decreasing wavelengths to avoid bleed-through and minimize photobleaching. The illumination and exposure settings for the acquisition of the gonad images were set up as previously described (John et al., 2025 preprint). Briefly, the *sygl-1* exon probe (TAMRA) and the *let-858* intron probe (TAMRA) were excited at 555 nm (40%) and the signals were acquired at 540-640 nm (gain was set to high well capacity) with an exposure time of 250 ms. The *sygl-1* intron probe (Quasar 670) was excited at 635 nm (40%) and the signal was acquired at 625-775 nm (gain was set to high well capacity) with an exposure time of 250 ms. DAPI was excited at 390 nm (10% illumination), and signal was acquired at 400-480 nm (gain high well capacity) with an exposure time of 50 ms. The images were

then processed through the THUNDER Imaging computational clearing method to reduce the excessive background signal generated from widefield microscopy.

**PZ extents**
In this study, the PZ extent was measured from the most distal end of the gonad to the end of the PZ, where a cell row with more than one crescent-shaped cell as previously described (Urman et al., 2023; John et al., 2025 preprint; Tolkin and Hubbard, 2021; Byrd et al., 2014; Crittenden et al., 2006; Fausett et al., 2023; Gordon, 2020; Roy et al., 2016).

**Notch *sygl-1* ATS and mRNA extents**
The *sygl-1* ATS extents and mRNA extents were measured as previously described to estimate the GSC pool size (Urman et al., 2023; John et al., 2025 preprint). The ATS extents were measured using the distance from the distal most end of the germline to the last ATS within the germline. The mRNA extents were measured using the distance from the distal most end of the germline to the end of the mRNA-rich region (<5 mRNA per cell) within the germline (Urman et al., 2023; John et al., 2025 preprint).

**Image processing using the custom-made MATLAB codes**
All processes were implemented and automated using modified MATLAB (v2.0) codes similar to the source code developed in our previous work (Crittenden et al., 2019; Lee et al., 2016; Urman et al., 2023; John et al., 2025 preprint; Lynch et al., 2022; John and Lee, 2025). After the analysis was completed, MATLAB and GraphPad Prism were used to visualize the data generated and conduct statistical tests as previously described (Urman et al., 2023; John et al., 2025 preprint).

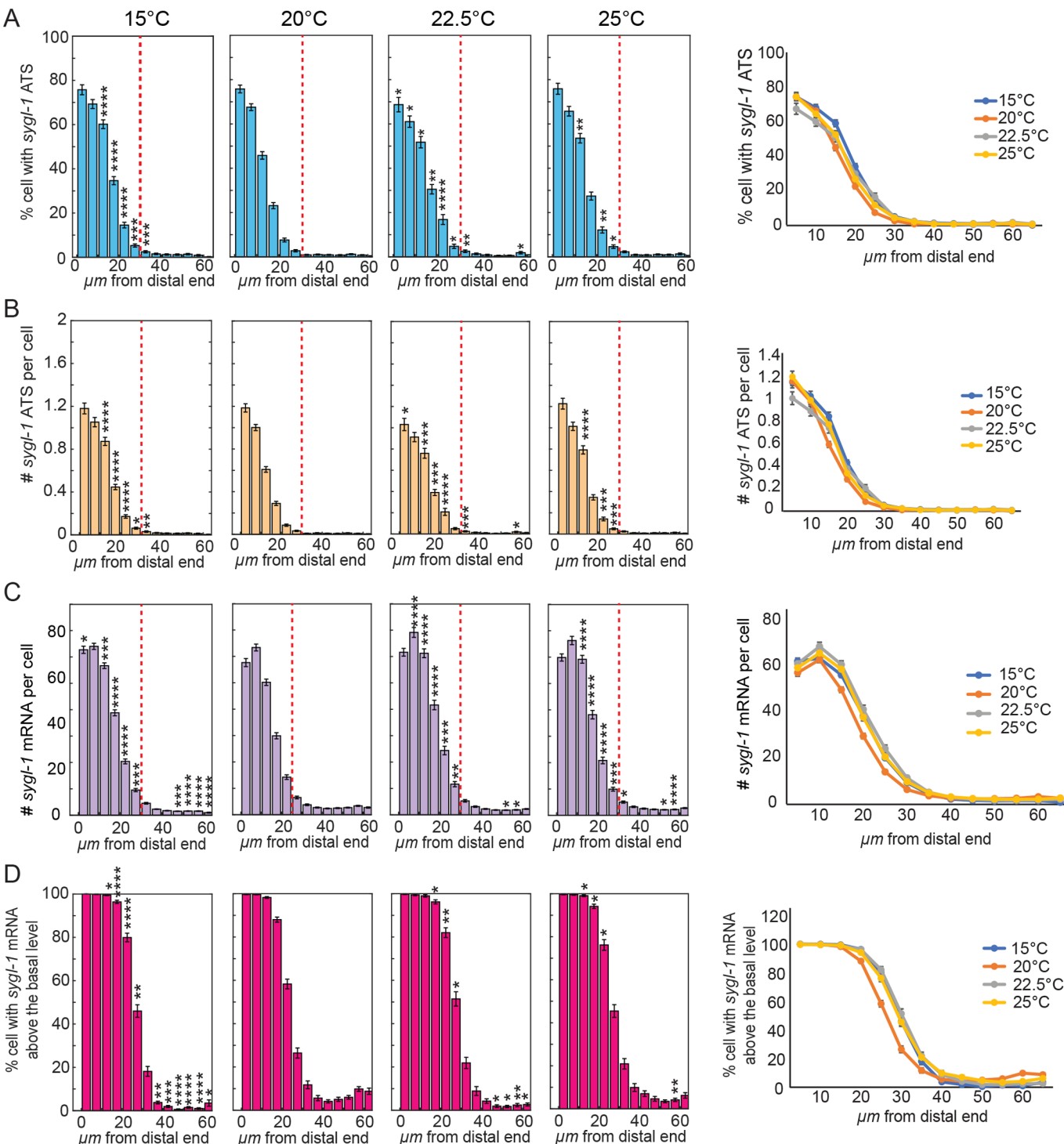

**Fig. 4. *sygl-1* transcriptional response is not affected spatially as temperature increases.** (A) Percentages of cells with *sygl-1* ATS were plotted against the function of position from the distal end in microns for N2 in varying temperatures. (B) Number of *sygl-1* ATS per cell was plotted against the function of position from the distal end in microns for N2 in varying temperatures. (C) The number of *sygl-1* cytoplasmic mRNA per cell was plotted against the function of position from the distal end in microns for N2 in the varying temperatures. (D) Percentages of cells with *sygl-1* cytoplasmic mRNA above the basal level were plotted against the function of position from the distal end in microns for N2 in varying temperatures. (A-D) Sample sizes for temperatures, 15°C, 20°C, 22.5°C and 25°C were as follows: *n*=86, 118, 63, and 84 gonads, respectively. *$P<0.05$; **$P<0.01$; ***$P<0.001$; ****$P<0.0001$ by two-tailed *t*-test.

If the datasets met the requirements for parametric statistical analysis through normality tests (Anderson-Darling normality test), ANOVA and *t*-tests were performed to compare datasets presented in this study. If the data set did not satisfy the requirements for parametric analysis, the Kolmogorov–Smirnov (KS) test (a nonparametric version of the *t*-test) was performed.

**Acknowledgements**
We are thankful for the resources provided by the Molecular Biology Core Facility in the Life Sciences Research Building and the RNA Institute at the University at Albany.)

**Competing interests**
The authors declare no competing or financial interests.

## Author contributions

Conceptualization: C.L.; Formal analysis: N.S.J., K.S.T., M.A.U.; Investigation: N.S.J., K.S.T., M.A.U.; Supervision: C.L.; Validation: C.L.; Visualization: N.S.J., M.A.U.; Writing – original draft: N.S.J.; Writing – review & editing: C.L.

## Funding

 Deposited in PMC for immediate release.

## Data and resource availability

All relevant data and details of resources can be found within the article and its supplementary information.

## Peer review history

The peer review history is available online at https://journals.biologists.com/bio/article-lookup/doi/10.1242/bio.062031.reviewer-comments.pdf.

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
