## [Peer Review File · Biology Open]

Comprehensive comparative analysis of the effects of temperature on the Notch signaling response in vivo

Nimmy S. John, Kah Seng Tang, Michelle A. Urman and ChangHwan Lee

DOI: 10.1242/bio.062031

Editor: Sandhya Koushika

Review timeline

Original submission:	24 April 2025
Editorial decision:	30 April 2025
First revision received:	8 September 2025
Accepted:	11 September 2025

Original submission

First decision letter

MS ID#: bio.062031

MS TITLE: Comprehensive comparative analysis of the effects of temperature on the Notch signaling response in vivo

AUTHORS: Nimmy S. John; Kah Seng Tang; Michelle A. Urman; ChangHwan Lee

I have now reached a decision on the above manuscript.

The reviewer reports are shown at the bottom of this email or can be accessed, together with a copy of this decision letter, by going to:

As you will see, the reviewers gave favourable reports, but raised some critical points that will require amendments to your manuscript. I hope that you will be able to carry these out, because we would like to be able to accept your paper.

At this stage, we also ask you to ensure your manuscript complies with our formatting guidelines “ please see our manuscript preparation guidelines for details. Provided you are able to fully address the referees’ comments, we are positive about publication of your paper (we accept over 95% of revision submissions) and therefore hope you won’t mind any extra work involved in reformatting your manuscript at this point.

Please ensure that you clearly highlight all changes made in the revised manuscript. Please avoid using 'Tracked changes' in Word files as these are lost in PDF conversion.

I should be grateful if you would also provide a point-by-point response detailing how you have dealt with the points raised by the reviewers in the 'Response to Reviewers' box. Please attend to all of the reviewers’ comments. If you do not agree with any of their criticisms or suggestions please explain clearly why this is so.

Reviewer 1

There are some inconsistencies between the text and the results presented:

- 1) It is stated that Notch activation remains robust across temperatures. However, in Figures 1D, S1A, and S1B, there are statistically significant differences observed at 25 °C. Please revise the text to accurately reflect these findings.
- 2) Similarly, regarding nuclear size (Figures 1F and S1C), the manuscript claims that nuclear size is essentially unchanged across temperatures. However, significant differences are detected at 22.5 °C and 25 °C. This discrepancy should be addressed and corrected in the text.
- 3) Was any statistical analysis performed for Figure 1E? To support the conclusion that there are no differences, a Chi-square test should be conducted.
- 4) Figure S1D is not referenced anywhere in the text. Please incorporate a reference and explanation for this figure.
- 5) In Figure 1G, it is stated that mitotically dividing cells increase at higher temperatures; however, this is not evident in the figure. Please clarify or adjust the interpretation accordingly.
- 6) The graphical styles across figures are inconsistent. I recommend homogenizing the styles for better readability and professionalism. Specifically, consider replacing violin plots with box-and-whisker plots (similar to Figure 2B) and, if possible, include individual data points.
- 7) Figure 1C has approximately half the sample size (n) compared to other figures. Please increase the sample size to match the statistical power of the other datasets, especially since visual inspection (Fig. 1A) suggests potential differences.
- 8) The analysis in Figure 2 appears less rigorous compared to Figure 1. The thickness differences shown in Figure 2A might result from variations in cell number. Please normalize your data to the number of cells. Additionally, it appears that multiple let-858 ATS are present within the same nuclei. Is there a difference in the number of ATS per ATS-containing cell, similar to the what was seen in Figure 1C? If so, please include this analysis.
- 9) The rationale for evaluating different distance intervals (e.g., 0-30, 0-20, 0-60) is unclear. Please justify the choice of these intervals.
- 10) The manuscript states that Notch activation and its spatial pattern are invariant across temperatures (lines 149-150). This is inaccurate, as changes are observed at 25 °C. Additionally, are the expression patterns of GLP-1, LAG-2, and APX-1 affected by temperature? Stability or variation in these factors could explain the changes observed in downstream targets like sygl-1. Please address this point.
- 11) Please specify in each figure legend which statistical tests were performed. While the Methods section mentions the use of t-tests, many of the comparisons involve more than two groups, where ANOVA would be more appropriate. If multiple t-tests were chosen instead of ANOVA, please provide a clear justification.
- 12) Figure 4, as presented, is not very informative. A more effective way to visualize these data would be through density plots or ridgeline plots, collapsing all temperatures into a single graph with different colors for each condition. This approach would better highlight distributional differences that are currently hinted at in the bar graphs but not discussed. Additionally, please perform Chi-square analyses to determine if the distributions differ significantly between temperatures.

Reviewer 2

Summary: The authors carry out a detailed analysis of sygl-1 gene expression, which likely reflects in vivo Notch signaling, at different temperatures using multiple assays that report Notch activity. The work suggests that the activation and spatial pattern of Notch signaling is insulated from temperature effects, however higher temperature might lead to higher Notch transcriptional output. The work will be of interest due to the focus of the work on a well-defined and well-studied Notch signaling process in the *C. elegans* germline and the use of standard growth temperatures. It is a strength that the authors are able to quantify at the chromosomal, single cell, and tissue level readouts of transcription using sygl-1 smFISH. The data acquired is multifaceted, robust, sensitive to subtle changes, and of high quality. The work provides novel and very strong foundational understanding of Notch signaling outputs that are likely going to promote further understanding of Notch signaling mechanisms.

1. Experimental quality
 - a. Does each figure have the proper controls?

No, in Figure 1 the authors need to show that the *sygl-1* probes being used are *glp-1* dependent.

No, in Figure 2 the authors need to show that nuclear number 30 microns from the distal tip does not increase at 25C leading to more *let-858* puncta at that temperature.

b. Are experiments performed using appropriate methods that will answer the question (or test the hypothesis or support the observations) posed by the authors? Is the right tool used for the job?

Yes, the authors use sensitive and fully appropriate tools to answer their experimental questions.

c. Were the data analyzed using appropriate statistical tests?

Yes in most cases, but as noted in my comments below, it is often unclear who is being compared in the statistical tests and the data in Figure 4 should be analyzed statistically.

2. Reproducibility

a. Were experiments in each figure performed using adequate number of biological replicates?

Yes!

b. Is there sufficient raw data to assess the rigor of the analysis?

Yes!

c. Does the methods section provide sufficient detail to permit reproducibility?

In most cases yes, if one does a lot of citation look up to fill in the methods, which are often sparse. As described below, there is some question about bleaching and imaging settings.

3. Completeness

a. Are the author's conclusions supported by the data?

In many cases yes. As discussed in the comments below, the amount of changes seen in the data in Figure 3 should be stated, and if they are very subtle, the authors should consider tempering their conclusions as to the effect of temperature on Notch transcriptional activity.

b. Are there any flaws in the experimental design that invalidate the approach taken by the authors?

No, the experiments are well designed and quantified!

c. Are there experiments that have not been performed, but if true would disprove the conclusion? If yes, and if such experiments would be costly or time-consuming to perform, do the authors acknowledge this in a discussion of the limitations?

As discussed in my comments below there are a few key controls that the authors should consider, and it is likely that they can be generated using existing data sets and MATLAB code.

4. Scholarship

a. Do the authors cite and discuss the merits of relevant data that would argue against their conclusion?

No.

b. Do the authors cite and discuss the merits of relevant data that would support their conclusion?

c. Yes.

Suggestions for improvement:

1. In the first paragraph of the introduction, it would be helpful to provide some mechanistic insights into the examples of how temperature can impact important developmental and physiological processes.

2. The rationale for the proposed experiments could be much be stronger. For example, the authors could describe why there is a (lines 46-47) "need for more direct and sensitive assays to assess how temperature modulates Notch activation and signaling dynamics".

3. The type of statistical tests performed should be described in each figure legend.

4. It would be helpful if the authors presented results showing that the *sygl-1* probes being used in this study report GLP-1/Notch activity.

5. It is unclear what the asterisks in Figure 1D and F refer to and who is being compared to each other with statistical tests in Figure 1D and E.

6. The authors make claims about differences between temperatures that appear to be inconsistent with the statistical analysis of the data (Figure 1F and G) in lines 89-93.

7. The authors present smFISH images in Figure 1A using the *sygl-1* exon probe, whose quantification is not presented until Figure 3. These images do not reflect the conclusions from the quantification in Figure 3, I suggest providing images that better align with the quantification in Figure 3.

8. It is unclear to me how the authors are interpreting the *sygl-1* intron puncta Figure 1A. Isn't it possible that a puncta represent one or many nascent transcripts? If this is the case then it would

not be reliable to use puncta number to accurately assess the amount of active transcription of *sygl-1*.

9. In Figure 2, it appears that the increased number of *let-858* intron probe puncta at 25C is due to a change in gonad morphology and an increase in nuclear number. This should be accounted for in the data analysis.

10. The data in Figure 2 is intended to be a control to compare to the data in Figure 1. However, the "Number of ATS 30 microns from the distal end" data shown in Figure 2 is not quantified or shown *vs* *sygl-1* intron puncta in Figure 1, significantly limiting the conclusions that can be drawn about the effects of 25C on these two transcripts. In addition, if there is an increase in the number of *let-858* intron puncta, it could be outside of the germline stem cell pool, where *sygl-1* transcription is restricted. This could be addressed by limiting the analysis of *let-858* intron puncta numbers to these cells.

11. It is unclear what the asterisks in Figure 2B refer to (p values?) and who is being compared to each other with statistical tests. Moreover, in Figure 2B a key to the box plot should be described, what do the horizontal line, box and whisker signify? Why are the authors not showing the raw data here as they do in Figure 1?

12. It is unclear what the asterisks in Figure 3 refer to (p values?) and who is being compared to each other with statistical tests. Moreover, while there appear to be statistically significant differences in some conditions and assays, it is often very difficult to tell from the graphs the extent of the differences. These should be provided in the results text, for example how much of a change in *sygl-1* intron puncta intensity and number per cell are seen between 20-25C or 15-25C? The amount of these changes are key in determining the strength of many of the authors' conclusions in the discussion.

13. The authors suggest that there is no change in the number of *sygl-1* intron puncta at different temperatures in their interpretations of the data presented in Figure 1. While this can be inferred from comparing the results in Figure 1B-E, I suggest that the authors show this quantification directly (as they do for *let-858* intron puncta in Figure 2B). These data seem important in interpreting the effects of temperature on *sygl-1* gene expression.

14. The authors present a very careful and informative spatial analysis of *sygl-1* intron puncta in Figure 4, this type of analysis with *let-858* intron puncta would be very useful in interpreting how temperature impacts its expression.

15. The authors suggest in the methods that there was no photobleaching, which seems unlikely. How did the authors account for this and how was imaging carried out to enable comparisons of *sygl-1* intron puncta intensity levels at various temperatures in Figure 3?

16. While the distributions of the bar graphs in Figure 4 look similar at different temperatures, have the authors carried out any statistical analysis of these to support their conclusions?

Minor comments:

1. Line 39 has a typo "Notch Signaling".

2. Lines 41-42: The citations 11-13 do not describe the stability of Notch signaling at different temperatures in *Drosophila*.

3. In line 64, please define ATS.

4. Are the horizontal lines in Figures 1 B-C the mean?

5. It would be helpful if the authors describe how the intron *vs* exon probes mark active transcription *vs* cellular mRNA levels.

6. In Fig 1A, what does the asterisk in 20-25C images near the distal tip of the gonad signify and what is the scale of the scale bar?

7. There is a typo in lines 218-219.

Reviewer's Responses to Questions

Experimental quality

Does each figure have the proper controls?

If 'No', please indicate reasons in Comments for Author box below.

Reviewer #1:

- Yes

Reviewer #2:

- No

Were the data analyzed using appropriate statistical tests?

If 'No', please indicate reasons in Comments for Author box below.

Reviewer #1:

- No

Reviewer #2:

- No

Reproducibility

Were experiments performed using adequate number of biological replicates?

If 'No', please indicate reasons in Comments for Author box below.

Reviewer #1:

- No

Reviewer #2:

- Yes

Does the methods section provide sufficient detail to permit reproducibility?

If 'No', please indicate reasons in Comments for Author box below.

Reviewer #1:

- Yes

Reviewer #2:

- No

Completeness

Are the manuscript's conclusions supported by the data?

If 'No', please indicate reasons in Comments for Author box below.

Reviewer #1:

- No

Reviewer #2:

- No

Scholarship

Do the authors cite and discuss the merits of data that would argue for and against their conclusion?

If 'No', please indicate reasons in Comments for Author box below.

Reviewer #1:

- Yes

Reviewer #2:

- No

Does the manuscript title & abstract accurately reflect the contents of the manuscript, without hyperbole?

If 'No', please indicate reasons in Comments for Author box below.

Reviewer #1:

- Yes

Reviewer #2:

- Yes

First revision

Author response to reviewers' comments

Comments from the Reviewers:

Reviewer 1: There are some inconsistencies between the text and the results presented:

1) It is stated that Notch activation remains robust across temperatures. However, in Figures 1D, S1A, and S1B, there are statistically significant differences observed at 25°C. Please revise the text to accurately reflect these findings.

We appreciate the reviewer's thoughtful feedback. In response, we have revised our manuscript as suggested (e.g., lines 106-112).

2) Similarly, regarding nuclear size (Figures 1F and S1C), the manuscript claims that nuclear size is essentially unchanged across temperatures. However, significant differences are detected at 22.5°C and 25°C. This discrepancy should be addressed and corrected in the text.

We have revised the manuscript to indicate the differences and elaborate on their statistical and biological significance (e.g., lines 112-117).

3) Was any statistical analysis performed for Figure 1E? To support the conclusion that there are no differences, a Chi-square test should be conducted.

We appreciate the reviewer's suggestion. We performed a Chi-square test with the data presented in Fig. 1E and have indicated the statistical significance in the figure. The text has been revised accordingly (e.g., lines 106-108).

4) Figure S1D is not referenced anywhere in the text. Please incorporate a reference and explanation for this figure.

We thank the reviewer for pointing this out. We have now incorporated a reference to Figure S1D in the Results section (e.g., lines 104-108) and added a brief explanation of the data to clarify its relevance.

5) In Figure 1G, it is stated that mitotically dividing cells increase at higher temperatures; however, this is not evident in the figure. Please clarify or adjust the interpretation accordingly.

We revised the texts (e.g., lines 115-117) so that they accurately reflect the data presented in Figure 1G.

6) The graphical styles across figures are inconsistent. I recommend homogenizing the styles for better readability and professionalism. Specifically, consider replacing violin plots with box-and-whisker plots (similar to Figure 2B) and, if possible, include individual data points.

We have revised all figures to display the data as violin plots, as the reviewer suggested (similar to Figure 2B). While we considered overlaying individual data points, the big sample size (hundreds to tens of thousands) obscured the plot, as the points essentially covered the entire violin. For clarity, we decided not to include them but are happy to add individual points if the reviewers still feel this would improve the presentation.

7) Figure 1C has approximately half the sample size (n) compared to other figures. Please increase the sample size to match the statistical power of the other datasets, especially since visual inspection (Fig. 1A) suggests potential differences.

We thank the reviewer for this suggestion. We have increased the sample size for Figure 1C and updated both the figure and the corresponding text.

8) The analysis in Figure 2 appears less rigorous compared to Figure 1. The thickness differences shown in Figure 2A might result from variations in cell number. Please normalize your data to the number of cells. Additionally, it appears that multiple *let-858* ATS are present within the same nuclei. Is there a difference in the number of ATS per ATS-containing cell, similar to the what was seen in Figure 1C? If so, please include this analysis.

We appreciate the reviewer's insightful suggestions. As recommended, we have normalized the *let-858* ATS numbers to cell number and now report these results in Figure 2D, with corresponding revisions made to the text. In addition, we performed the spatial analysis analogous to Figure 4, which is now presented in Figures 2E and 2F. The majority of germ cell nuclei contained either zero or one ATS, and this remained consistent across temperatures, in agreement with Figures 2D-F.

9) The rationale for evaluating different distance intervals (e.g., 0-30, 0-20, 0-60) is unclear. Please justify the choice of these intervals.

We have added justification and clarification for using different intervals to the text (e.g., lines 106-108).

10) The manuscript states that Notch activation and its spatial pattern are invariant across temperatures (lines 149-150). This is inaccurate, as changes are observed at 25°C. Additionally, are the expression patterns of *GLP-1*, *LAG-2*, and *APX-1* affected by temperature? Stability or variation in these factors could explain the changes observed in downstream targets like *sygl-1*. Please address this point.

We have revised the text to indicate the changes at 25°C, although we note that ATS-level changes are minimal (Figures 1-5 and S1, typically <2%). Data analyzing *GLP-1* and *LAG-2* at the RNA and protein levels across temperatures and developmental stages are included in a separate manuscript currently under revision. We are also in the process of generating a new strain expressing epitope-tagged endogenous *apx-1*, but our initial attempts have not yet been successful.

11) Please specify in each figure legend which statistical tests were performed. While the Methods section mentions the use of t-tests, many of the comparisons involve more than two groups, where ANOVA would be more appropriate. If multiple t-tests were chosen instead of ANOVA, please provide a clear justification.

We appreciate the reviewer's suggestion. Following standard statistical procedures, we first performed one-way ANOVA to assess overall differences among multiple groups (reported by F-values). For pairwise comparisons, we applied two-tailed t-tests, as this approach is appropriate when comparing two populations among several that share similar variances, as confirmed by the ANOVA results. We have updated the figure legends, Results text, and Methods section to clearly indicate which statistical tests were used and provide justification for using pairwise t-tests following ANOVA where applicable.

12) Figure 4, as presented, is not very informative. A more effective way to visualize these data would be through density plots or ridgeline plots, collapsing all temperatures into a single graph with different colors for each condition. This approach would better highlight distributional differences that are currently hinted at in the bar graphs but not discussed. Additionally, please perform Chi-square analyses to determine if the distributions differ significantly between temperatures.

We thank the reviewer for this insightful suggestion to improve data visualization. In response, Figures 2 and 4 have been revised to include overlapping ridgeline plots, effectively highlighting similarities and differences in ATS spatial distributions across temperatures. Regarding statistical analysis, we have retained pairwise t-tests for these figures, as they provide the most direct comparison of ATS probabilities at corresponding gonadal positions. While a multi-populational Chi-square test assesses overall distribution homogeneity, it does not capture the location-specific comparisons critical to our conclusions. Nonetheless, we are happy to provide Chi-square results if the reviewers feel this would add further context.

Reviewer 2: Summary: The authors carry out a detailed analysis of *sygl-1* gene expression, which likely reflects in vivo Notch signaling, at different temperatures using multiple assays that report Notch activity. The work suggests that the activation and spatial pattern of Notch signaling is insulated from temperature effects, however higher temperature might lead to higher Notch transcriptional output. The work will be of interest due to the focus of the work on a well-defined and well-studied Notch signaling process in the *C. elegans* germline and the use of standard growth temperatures. It is a strength that the authors are able to quantify at the chromosomal, single cell, and tissue level readouts of transcription using *sygl-1* smFISH. The data acquired is multifaceted, robust, sensitive to subtle changes, and of high quality. The work provides novel and very strong foundational understanding of Notch signaling outputs that are likely going to promote further understanding of Notch signaling mechanisms.

1. Experimental quality

a. Does each figure have the proper controls?

No, in Figure 1 the authors need to show that the *sygl-1* probes being used are *glp-1* dependent.

We appreciate the reviewer's comment. The *sygl-1* smFISH probes used in this study were originally developed by Lee et al. (2016, eLife) and have been extensively validated for their specificity and *glp-1* dependence in multiple publications, including our recent work (Urman et al., 2024, BiO). To address the reviewer's comment, we have revised the Introduction to explicitly note their specificity with supporting references.

No, in Figure 2 the authors need to show that nuclear number 30 microns from the distal tip does not increase at 25C leading to more *let-858* puncta at that temperature.

To address this concern, we added new analyses presented in Figures 2C and 2D and revised the text accordingly.

b. Are experiments performed using appropriate methods that will answer the question (or test the hypothesis or support the observations) posed by the authors? Is the right tool used for the job?

Yes, the authors use sensitive and fully appropriate tools to answer their experimental questions.

c. Were the data analyzed using appropriate statistical tests?

Yes in most cases, but as noted in my comments below, it is often unclear who is being compared in the statistical tests and the data in Figure 4 should be analyzed statistically.

2. Reproducibility

a. Were experiments in each figure performed using adequate number of biological replicates?

Yes!

b. Is there sufficient raw data to assess the rigor of the analysis?

Yes!

c. Does the methods section provide sufficient detail to permit reproducibility?

In most cases yes, if one does a lot of citation look up to fill in the methods, which are often sparse. As described below, there is some question about bleaching and imaging settings.

We have added more details to the Methods section as suggested by the reviewer below.

3. Completeness

a. Are the author's conclusions supported by the data?

In many cases yes. As discussed in the comments below, the amount of changes seen in the data in Figure 3 should be stated, and if they are very subtle, the authors should consider tempering their conclusions as to the effect of temperature on Notch transcriptional activity.

b. Are there any flaws in the experimental design that invalidate the approach taken by the authors?

No, the experiments are well designed and quantified!

c. Are there experiments that have not been performed, but if true would disprove the conclusion? If yes, and if such experiments would be costly or time-consuming to perform, do the authors acknowledge this in a discussion of the limitations?

As discussed in my comments below there are a few key controls that the authors should consider, and it is likely that they can be generated using existing data sets and MATLAB code.

We have addressed the reviewer's comments below.

4. Scholarship

a. Do the authors cite and discuss the merits of relevant data that would argue against their conclusion?

No.

b. Do the authors cite and discuss the merits of relevant data that would support their conclusion?

c. Yes.

Suggestions for improvement:

1. In the first paragraph of the introduction, it would be helpful to provide some mechanistic insights into the examples of how temperature can impact important developmental and physiological processes.

We thank the reviewer for this suggestion. We have revised the Introduction accordingly.

2. The rationale for the proposed experiments could be much stronger. For example, the authors could describe why there is a (lines 46-47) "need for more direct and sensitive assays to assess how temperature modulates Notch activation and signaling dynamics".

We revised the sentence as suggested.

3. The type of statistical tests performed should be described in each figure legend.

We have revised figure legends to specify the statistical tests used in each analysis.

4. It would be helpful if the authors presented results showing that the *sygl-1* probes being used in this study report GLP-1/Notch activity.

As mentioned above, we have revised the Introduction with relevant references to show the specificity and GLP-1 dependence of the probes.

5. It is unclear what the asterisks in Figure 1D and F refer to and who is being compared to each other with statistical tests in Figure 1D and E.

The asterisks indicate the statistical significance based on pairwise two-tailed t-tests of the data (e.g., # ATS at 25°C) compared to the standard 20°C, unless otherwise specified. We have revised figure legends to explicitly state the meaning of the asterisks and the basis of comparison.

6. The authors make claims about differences between temperatures that appear to be inconsistent with the statistical analysis of the data (Figure 1F and G) in lines 89-93.

As the first reviewer also suggested, we have revised the text to explicitly describe the similarities and differences in *sygl-1* activity across temperature, consistent with the data presented in the figures.

7. The authors present smFISH images in Figure 1A using the *sygl-1* exon probe, whose quantification is not presented until Figure 3. These images do not reflect the conclusions from the quantification in Figure 3, I suggest providing images that better align with the quantification in Figure 3.

We have updated the images in Figure 1A to better align with and represent the quantifications presented in Figure 3.

8. It is unclear to me how the authors are interpreting the *sygl-1* intron puncta Figure 1A. Isn't it possible that a puncta represent one or many nascent transcripts? If this is the case then it would not be reliable to use puncta number to accurately assess the amount of active transcription of *sygl-1*.

We appreciate the reviewer's comment. The number of ATS (intron puncta) reflects the number of *sygl-1* loci actively engaged in transcription, representing the Notch transcriptional response at the chromosomal and cellular levels (e.g., Figure 1E). In contrast, ATS signal intensity (brightness/size) reflects the number of *sygl-1* nascent transcripts generated at each locus (e.g., Figure 3A). Counting ATS number has already been well established and validated as a reliable Notch readout for assessing transcriptional activation pattern at the cellular level (e.g., Lee et al., eLife, 2016). We have revised the text to clarify the interpretation of these assays.

9. In Figure 2, it appears that the increased number of *let-858* intron probe puncta at 25C is due to a change in gonad morphology and an increase in nuclear number. This should be accounted for in the data analysis.

To address the reviewer's comment, we added new analyses in Figures 2C-D, which confirm that *let-858* transcription levels are not affected by temperature.

10. The data in Figure 2 is intended to be a control to compare to the data in Figure 1. However, the "Number of ATS 30 microns from the distal end" data shown in Figure 2 is not quantified or shown doe *sygl-1* intron puncta in Figure 1, significantly limiting the conclusions that can be drawn about the effects of 25C on these two transcripts. In addition, if there is an increase in the number of *let-858* intron puncta, it could be outside of the germline stem cell pool, where *sygl-1* transcription is restricted. This could be addressed by limiting the analysis of *let-858* intron puncta numbers to these cells.

We appreciate the reviewer's comment. To address this concern, we performed a new spatial analysis of *let-858*-containing cells across the distal germline (0-60 μm), now presented in Figures 2E-F, which demonstrates the uniform distribution of ATS. The text has been revised accordingly.

11. It is unclear what the asterisks in Figure 2B refer to (p values?) and who is being compared to each other with statistical tests. Moreover, in Figure 2B a key to the box plot should be described, what do the horizontal line, box and whisker signify? Why are the authors not showing the raw data here as they do in Figure 1?

As mentioned above, all plots have been converted to violin plots where applicable. We also revised figure legends to clearly indicate the meaning of the asterisks and specify that they represent comparison to the standard 20°C.

12. It is unclear what the asterisks in Figure 3 refer to (p values?) and who is being compared to each other with statistical tests. Moreover, while there appear to be statistically significant differences in some conditions and assays, it is often very difficult to tell from the graphs the extent of the differences. These should be provided in the results text, for example how much of a change in *sygl-1* intron puncta intensity and number per cell are seen between 20-25C or 15-25C? The amount of these changes are key in determining the strength of many of the authors conclusions in the discussion.

We thank the reviewer for this helpful comment. As mentioned above, we have revised the figure legends to clearly indicate both the meaning of the asterisks and the basis of comparison. We also updated the text to indicate percentage changes where appropriate and informative. In addition,

the newly added ridgeline plots with overlapping spatial patterns (Figures 2F and 4) provide a clearer visualization of transcriptional activation patterns across temperatures within the same plot, enabling direct comparison.

13. The authors suggest that there is no change in the number of *sygl-1* intron puncta at different temperatures in their interpretations of the data presented in Figure 1. While this can be inferred from comparing the results in Figure 1B-E, I suggest that the authors show this quantification directly (as they do for *let-858* intron puncta in Figure 2B). These data seem important in interpreting the effects of temperature on *sygl-1* gene expression.

The data were already presented in Figure S1A-D, and we have revised the text to clearly reference and explain these results.

14. The authors present a very careful and informative spatial analysis of *sygl-1* intron puncta in Figure 4, this type of analysis with *let-858* intron puncta would be very useful in interpreting how temperature impacts its expression.

As mentioned above in response to a similar concern from Reviewer 1, we expanded our analyses of *let-858* and incorporated them into Figure 2.

15. The authors suggest in the methods that there was no photobleaching, which seems unlikely. How did the authors account for this and how was imaging carried out to enable comparisons of *sygl-1* intron puncta intensity levels at various temperatures in Figure 3?

We appreciate the reviewer's comment. We revised the sentence and add more details to avoid any confusion. Our MATLAB analysis codes account for photobleaching by applying multiple normalization steps, using background signals both inside and outside the germline and corresponding nucleus in each focal plane. We have revised the Methods section to provide additional details and included relevant references, including our recent publication that describes this methodology.

16. While the distributions of the bar graphs in Figure 4 look similar at different temperatures, have the authors carried out any statistical analysis of these to support their conclusions?

We performed pairwise two-tailed t-tests to assess statistical significance, which is indicated with asterisks and explained in the figure legends. We also added new ridgeline plots, as mentioned above, for a clearer visualization of spatial patterns within a single plot (Figures 2 and 4).

Minor comments:

1. Line 39 has a typo "Notch Signaling".

We appreciate the reviewer's comment and have corrected the typo accordingly.

2. Lines 41-42: The citations 11-13 do not describe the stability of Notch signaling at different temperatures in *Drosophila*.

The citations have been updated to reference the correct publications.

3. In line 64, please define ATS.

We have added the definition of ATS in the line 64.

4. Are the horizontal lines in Figures 1 B-C the mean?

They indicate the medians, and we have revised the figure legends to clearly state this.

5. It would be helpful if the authors describe how the intron vs exon probes mark active transcription vs cellular mRNA levels.

We have revised the text and Methods section to clearly explain this along with relevant references.

6. In Fig 1A, what does the asterisk in 20-25C mages near the distal tip of the gonad signify and shat is the scale of the scale bar?

We have added explanations of the statistical meaning of the asterisks and included scale bar information in the figure legends.

7. There is a typo in lines 218-219.

We have corrected the typo.

Second decision letter

MS ID#: bio.062031R1

MS TITLE: Comprehensive comparative analysis of the effects of temperature on the Notch signaling response in vivo

AUTHORS: Nimmy S. John; Kah Seng Tang; Michelle A. Urman; ChangHwan Lee

I am happy to tell you that your manuscript has been accepted for publication in Biology Open, pending our standard publication integrity checks. It was accepted on 11th September 2025.